# Research on Risk and Resilience Evaluation of Urban Underground Public Space

**DOI:** 10.3390/ijerph192315897

**Published:** 2022-11-29

**Authors:** Xiaojuan Li, Lulu Li, Mingchao Lin, Chi Yung Jim

**Affiliations:** 1College of Transportation and Civil Engineering, Fujian Agriculture and Forestry University, Fuzhou 350108, China; 2Department of Social Sciences, Education University of Hong Kong, Lo Ping Road, Tai Po, Hong Kong, China

**Keywords:** underground public space, safety science, risk assessment, resilience evaluation, resilience improvement

## Abstract

High urban density, land scarcity, rapid population growth, and traffic congestion have restricted urban development. In response, selected multiple functions have increasingly been integrated into the underground public space (UPS) to maximize the 3D utilization of precious urban space. The accelerated intensity of UPS use has alerted safety concerns. UPS with enclosed and confined natures, complex building structures, locations usually in cramped areas, and limited emergency exits are potentially more prone to heavy casualties and losses in natural or human-made disasters. As research on UPS safety is limited and focused on single risks, we attempted to fill the knowledge gap by developing an integrated risk analysis of UPS to understand risk resilience and improve risk management. From the perspective of the UPS system, four latent factors were identified: natural environment, economic environment, facilities and equipment, and physical structure. Seventeen resilience indicators subsumed under the factors were selected based on resilience concepts. A questionnaire was designed to gather opinions on the relative importance rating of the resilience indicators. SPSS and AMOS software were enlisted to build a structural equation model (SEM), validate the data and model, and calculate the path coefficients and index weights to test four hypotheses. The SEM model results were employed to develop a holistic resilience enhancement strategy under a four-phase framework: before, during, after, and long-term, and under four latent factors. The resilience enhancements can optimize UPS disaster prevention, rescue and evacuation, mitigation, and response management.

## 1. Introduction

Urban development in high-density areas often necessitates using underground spaces. They include mainly subways (underground railways) and underground shopping venues. The increasingly heavy use of such underground public space (UPS) has engendered common and acute safety problems [1,2]. Compared with above ground public spaces, UPS is more vulnerable to various disasters, which are more difficult to prevent and less able to recover afterward [3].

UPS usually has a complex system with unique internal and inter facial traits requiring different safety management. They include enclosed space, built structures densely packed in a limited volume of space, diverse and closely juxtaposed functions, complicated networks of pedestrian passages with multiple nodes and segments, considerable vertical distances between some levels, crowded and highly mobile people flow, and a limited number of entrances and exits [4,5]. Many potential risks could trigger safety mishaps in UPS. Accidents such as escalator failure or power failure are likely to cause mass panic and are prone to congestion and catastrophic stampede. Such unfortunate accidents could cause mass crush injuries and death and considerable economic losses [6,7]. 

The safety concerns in UPS with the potential to develop into mass casualty incidents demand meticulous management and research on risks and their prevention. The critical concerns are identifying and evaluating the risks, developing suitable preventive measures and emergency responses, and striking a balance between operation and regulation [8,9]. Due to a relatively high degree of uncertainty and unpredictability of the risks, the traditional concept of disaster prevention may not suit UPS. The concept of resilience can provide novel ideas for urban safety administration from the dynamic perspective [10]. It can be proactively applied, focusing on the response to disasters as well as emphasizing the ability to adapt to disasters [11].

The application of the resilience theory [12] to underground spaces has received little research attention. However, the limited findings have been adopted to improve UPS’s ability to prevent and mitigate disasters. Resilience can be understood as the ability to respond promptly and effectively to a risk event and to recover quickly the normal operating functions. Dai et al. [13] studied the impact of underground space on the disaster prevention capacity of cities. Zou et al. [14] analyzed the shortcomings of urban underground space layout design and spatial resilience characteristics from the concept of disaster prevention and the resilient city, and then proposed a new special strategy for disaster prevention and mitigation in underground space. Ferreira et al. [15] proposed a strategy for building, maintaining, and managing green infrastructure to improve the resilience of urban ecosystems. Esposito et al. [16] studied mitigating the negative impacts of subways from disaster events by applying individual asset vulnerability indicators and system optimization models to improve the resilience of underground rail transport. Zhang et al. [17] used the CRITIC-entropy and TOPSIS methods to construct an evaluation model of urban resilience in three-dimensional space. They explored the shortcomings of building resilience in the physical, social, and information spaces of individual cities, and proposed strategies to enhance urban resilience.

A resilience evaluation system and evaluation indicators can provide the basis for quantification and implementation. Xu et al. [18] developed a resilience evaluation index system for UPS with six dimensions: government regulation, physical structure, dense nodes of passenger flow, natural environment, traffic accessibility, and economic environment. Sun et al. [19] constructed an index system from four perspectives of social development, economy, ecology, and municipal facilities, and conducted an evaluation study on the resilience value and resilience state of cities; Huang et al. [20] explored the application of the resilience theory in the field of security science. They designed a system security resilience framework and identified the role of system security in resilience research. Li [21] reviewed articles on resilient cities in major international journals from 2000 to 2015. They found that the index system used for resilient city evaluation covered social, economic, institutional, and infrastructure resilience. Xie et al. [22] used hierarchical analysis to calculate the weights of indicators from four dimensions: economic level, leadership, infrastructure, and ecological environment, and used the fuzzy integrated decision-making method to analyze the resilience of cities.

Most research applied the resilience theory to urban development, rail transportation, and other safety fields [23,24]. The UPS system has received little research attention, which calls for specific studies on risks in combination and their spatial coupling. In view of the knowledge gaps, this study aimed to improve the safety of UPS by analyzing the potential hazards and proposing solutions and improvements. We evaluated the concept, types, and general characteristics of UPS, taking into account the current situation and future trends in China and other countries. After examining the safety hazards and risk events, we identified the risk factors, conducted risk assessment, and proposed resilience enhancement strategies in line with the resilience theory.

Therefore, the purpose of this study is to identify and analyze the possible risk factors of UPS, establish a model to measure the risk level of UPS, and propose resilience enhancement strategies for UPS in combination with resilience theory to improve the safety of UPS. The focus of the next section is to identify the main risk factors of the UPS based on the literature.

We consulted the risk management processes in the ISO 31000:2018 Risk Management Standard [25]. The dynamic changes in resilience were considered based on the development and evolution of risk events. System-level risk response measures were proposed from the perspective of resilience to enhance the system’s ability to respond promptly and resume its normal operational functions quickly after a shock. We then developed a system resilience framework for evaluation and enhancement from four dimensions: natural environment, economic environment, social environment, and physical structure. The structural equation model (SEM) was enlisted to evaluate the resilience of UPS systems, including identifying the key factors and conducting an overall assessment. The system resilience was considered in four stages: before, during, after, and long-term development. Based on the resilience evaluation, we recommended strategies for UPS system resilience enhancement, which could provide a theoretical and practical basis for urban risk management.

## 2. Factors Shaping the Resilience of the UPS System

The risks of the UPS system are different from above ground public spaces due to a limited space environment and a certain degree of confinement. It can be affected by economic, environmental, management, technical, organizational, and other factors, with a bearing on safety accidents and human and economic losses [26]. Human losses include casualties of passengers, consumers, and staff, and economic losses include damage to subway trains, malfunction or damage to mechanical equipment, and destruction of building structures and property.

Resilience is an inherent ability of a system to resist and adapt to external risks. Based on the studies of disasters in UPS, the system resilience can be defined as the ability to continuously and dynamically adjust and resist external disturbances and ensure the normal operation of functions soon after the disruptive event [27,28]. When an unexpected stressful event occurs, a less resilient system will return to its usual stable level at a sluggish pace. Conversely, a more resilient system will react quickly to the unexpected event, enabling the system to return to a stable normal state as soon as possible to ensure system stability and equilibrium [29,30].

The study started from the perspective of system resilience and conducted response research at the system level concerning risk events in UPS. The resilience analysis covered four dimensions: natural environment, economic environment, facilities and equipment, and physical structure. The selection of these four latent factors was based on the Design Code for the Internal Environment of Urban Underground Space [31] and related literature [32,33]. Under each latent factor, the resilience indicators are identified (Table 1).

(1) Natural environment factor

Natural disasters are important UPS risk factors, so eliminating or responding to the effects brought by them can improve the safety level [34,35]. The natural environment factors concerning the resilience of the UPS system are listed in Table 1.

(2) Economic environment factor

From the perspective of the economic environment, the impacts of safety accidents on UPS cover three main aspects: prevention before the risk event, support when the risk occurs, and recovery after the risk [31]. Therefore, the economic factors with a bearing on UPS system resilience are related mainly to the resources, preparedness, capacity, capability, timeliness, and promptness in realizing the emergency plan. The key resilience indicators [36] are given in Table 1. 

(3) Facilities and equipment factor

The effectiveness of facilities and equipment furnishes the material basis of UPS. Based on previous studies, the four indicators shaping UPS resilience are included in Table 1 [37,38]. They play an indispensable role in the system’s daily operation. Disruption of suspension of any one indicator may lead to safety accidents. The effectiveness and capability and effectiveness of the facilities and equipment have a significant bearing on the ability to cope with disasters. 

(4) Physical structure factor

The physical structure’s functional integrity constitutes the foundation for UPS’s safety and security and an essential prerequisite to preventing and tackling risk events. The UPS system is more vulnerable to disaster impacts if the physical structure is unstable or poorly configured, which can increase the probability of risk and severity of its consequences [34]. The relevant resilience indicators are included in Table 1.

## 3. Methods

### 3.1. SEM Construction and Assumptions

The SEM model for UPS system resilience was divided into the measurements of sub-model and structural sub-model. They were then combined to build the complete SEM (Figure 1). 

Based on the findings in the literature on the resilience of urban underground spaces, four hypotheses were developed corresponding to the four latent factors: natural environment, economic environment, facilities and equipment, and physical structure (Table 2).

### 3.2. Questionnaire Design and Survey

The questionnaire survey method was applied to evaluate the resilience of the UPS system. The questionnaire contains two parts (Appendix A). The first part collected basic socio-economic information of the respondent, including education level, work experience, and occupation. The second part assessed the importance rating of the 17 resilience indicators. The scores were based on a 5-point bipolar Likert scale and their assigned numerical scores are: “Very important” (5), “Important” (4), “Average” (3), “Unimportant” (2), and “Very unimportant” (1). 

Four kinds of respondents were targeted: (a) the professionals in the construction and transportation industries; (b) pedestrians in the shopping venues and subway train passengers; (c) students and faculty members in the College of Architecture and Transportation; and (d) staff of the commercial concerns and subway railway staff in the UPS. 

### 3.3. Basic Statistical Analysis

Of the 250 distributed questionnaires including for living laboratories, 224 were collected, denoting a response rate of 89.6%. They were scrutinized to exclude those with wrong or missing answers, leaving 208 valid questionnaires representing a success rate of 92.8%, which met the SEM sample-size requirement [39]. The SPSS 26.0 software was used for data analysis.

Table 3 summarizes the socio-economic profile of the respondents. Most of them were well educated from technical colleges to bachelor and master’s degrees and above. Respondents with 3–5 years of work experience took up 38.0%, followed by 26.9% with 1–2 years and 22.1% with 6–10 years. By occupation, the construction industry is dominant at 34.6%, followed by 17.3% in transportation and 10.6% in corporations. 

Table 4 summarizes the importance rating of the 17 resilience indicators. Of the four groups of latent factors, facilities and equipment had the highest average rating value of 3.74, exceeding the overall average of 3.33 by a notable margin. The four top ranks 1–4 were all subsumed under this group. The economic environment is the second-ranked group with an average rating value of 3.56. It included ranks 5–8, and 11. Physical structure with a value of 3.36 was close to the overall average. It included ranks 9, 10, 12 and 13. The natural environment received the lower average value of merely 2.61. It took up the lowest ranks (14–17). 

### 3.4. Data Quality Analysis

(1) Running reliability analysis

The reliability of the questionnaire data can reflect the validity of the measured factors [40]. With similar conditions, the same event can be repeated or re-occur. Reliability analysis is the reflection of the consistency, stability, and repeatability of the results. The range of reliability coefficients is [0, 1], and a high Cronbach’s alpha coefficient indicates a more reliable test result. The reliability coefficient was divided into five bipolar grades (Table 5).

The reliability analysis was carried out by the alpha reliability coefficient method (Cronbach’s alpha reliability). Common to questionnaire analyses, the alpha coefficient was calculated by the SPSS 26.0 software to obtain Cronbach’s alpha coefficients (Table 6). The alpha coefficients of the four latent factors exceeded 0.7, indicating high reliability. Deleting questions did not increase alpha, indicating that the questionnaire results had good internal consistency.

(2) Conducting structural validity analysis

The questionnaire data were subjected to a validity analysis to detect the correlation between the sample and actual data. A strong correlation denotes high validity. The validation factor analysis was used to assess the structural validity of the model. The results were obtained by SPSS statistical data calculation, analysis-dimensional reduction-factor analysis, and calculation. Principal components were extracted employing the maximum variance rotation. The KMO appropriateness parameter test and Bartlett’s sphericity test determined the validity, indicating that the sample was suitable for factor analysis.The KMO value range and significance grading scale are shown in Table 7.

To further examine the reliability of the grading scale, the bias coefficient was added, which is the coefficient obtained by eliminating one of the measured items. The overall reliability can be improved by excluding items that do not meet the requirements. The validity analysis of the resilience indicators was calculated by SPSS using the analysis-dimensional reduction-factor program, and the results are shown in Table 8. The KMO value of 0.831 exceeded the minimum acceptable level of 0.5, indicating suitability for factor analysis. In addition, Bartlett’s sphericity test yielded significance level of *p* < 0.001, further signifying that the data were suitable for factor analysis.

(3) Performing convergent validity analysis

The convergent validity analysis can measure whether the questionnaire results correspond to the latent factors. The convergent validity analysis can measure whether the questionnaire results correspond to the latent factors. The validation factor analysis was conducted by the AMOS 23.0 software using the confirmatory factor analysis (CFA). The scale was analyzed for convergent validity using combined reliability (CR) and average variance extracted (AVE). The test results are shown in Table 9. The model’s standardized factor loadings of the 17 resilience indicators ranged from 0.599 to 0.887. They were all > 0.5, indicating that the latent factors were highly representative. The AVE extracted from the latent factors ranged from 0.560 to 0.730, all > 0.5. The CR values ranged from 0.831 to 0.915, all > 0.7. These results indicated acceptance of the designed convergent validity of the latent factors.

### 3.5. SEM-Based Path Analysis of the Resilience Model for UPS System

#### 3.5.1. Establishing Structural Equation Model

The maximum likelihood estimation method of the structural equation model was applied to the questionnaire data using AMOS. The method tested the significance of the path coefficients calculated by the model. The path diagram and path coefficients of the first-order structural equation model were calculated (Figure 2). The model parameters were estimated and presented in Table 10.

Figure 2 indicates that the measurement model had no negative differences. Therefore, it passed the model violation estimation check to indicate model validity. Table 10 shows that the standardized coefficient of A2 intensity of natural disasters (under natural environment) is 1.436, the CR is 9.038, and the SE values are greater and close to 0. The significance level is *p* < 0.05, indicating that the model is fit. Based on the theoretical and UPS first-order CFA model estimation results (Figure 2), the four latent variables are highly correlated with each other (correlation coefficients range from 0.15 to 0.31), suggesting that the model may have a higher-level factor structure, so this paper constructs a second-order CFA method to validate the measurement model. The second-order CFA model of UPS was also constructed using AMOS software (Figure 3).

#### 3.5.2. Conducting Model Fitting Test

Three composite indexes, namely absolute fit index, relative fit index, and parsimonious fit index, were selected to re-test the fitted measurement model. They tested whether the model fitted the data. The test does not need to satisfy all the fitting indexes, so only χ^2^, df, χ^2^/df, NFI, IFI, TLI, CFI, GFI, and RMSEA need to be satisfied. The results are shown in Table 11.

The questionnaire data were imported into the AMOS software. The model fitting parameters obtained by applying the maximum likelihood method are listed in Table 11. The fitted parameters all met the threshold criteria, indicating that the model fitted the data well. 

#### 3.5.3. Path analysis of Second-Order Structural Equation Model

(1) Running path analysis

To further explore the influence of resilience indicators on the UPS system, a second-order structural equation model was constructed in AMOS. The computed path coefficients are given in Figure 3.

(2) Conducting model fitting test

The results of parameter estimation for the second-order structural equation model and fitness test are compiled in Table 12.

The overall fitness calculations of the second-order structural equation model for UPS system resilience are listed in Table 13. The fitting values met the significance requirements; thus, the model did not require revision. The results could be used to test the model’s hypotheses.

We adopted Cohen’s criteria to judge the effect of path coefficients [42]. A value > 0.3 has a strong effect, 0.2–0.3 a moderate effect, and < 0.2 a weak effect. Figure 3 shows that the standardized path coefficients of all resilience indicators were >0.3 and positive, lying between 0.61 and 0.87. Table 12 shows that all SE values were >0 with no negative values, all CR values were >1.96, and all significance levels were *p* < 0.05. These results satisfied the requirements of statistical significance and acceptance.

#### 3.5.4. Calculating Risk Impact Weights

Table 12 shows that the computed path coefficients of the four UPS latent factors are 0.392, 0.613, 0.406, and 0.520, with normalized values of 20.3%, 31.7%, 21.0%, and 26.9%. They showed that economic environment and physical structure were more influential than the remaining latent factors. The normalized weights and total normalized weights are shown (Table 14).

#### 3.5.5. Hypothesis Testing and Resilience Analysis

The four hypotheses paraphrased in Table 2 could be tested by the modeling results shown in Table 14. The positive and relatively high path coefficients indicated that the hypotheses could be supported. Therefore, the four latent factors and their 17 resilience indicators positively affect UPS’s resilience. The path coefficient values offer a quantitative index of the relative contributions of the individual latent factors and resilience indicators. The latent factors could be arranged in descending order of importance: economic environment (0.613) > physical structure (0.520) > facilities and equipment (0.406) > natural environment (0.392). The human and artificial domains prevailed over nature’s attributes. Eleven of the seventeen resilience indicators had a path coefficient > 0.800, implying their strong positive impacts on UPS resilience. The most important contributors were the electricity supply system (0.885) and financial allocation for post-disaster relief (0.871). The least important were flood control and drainage system (0.600) and ventilation system (0.599).

## 4. Resilience Enhancement Strategies for UPS Systems

### 4.1. Developing the Resilience Strategy 

Most research in China on urban disaster prevention and mitigation focused on the aboveground urban public space, with few on the UPS. Moreover, most studies investigated a single-risk type, such as fire and flood, with few exploring the system resilience and disaster management from a holistic perspective. 

UPS system resilience is principally a time-dimensional issue. The core consideration is improving the resilience level to implement the whole-cycle management. It ranges from pre-disaster prevention and control, to mid-disaster emergency response, post-disaster recovery and reconstruction, and long-term development of UPS resilience. The full spectrum of resilience enhancement strategy of UPS systems can be assessed in four stages (ex-ante, in-event, post-event, and long-term development) with reference to four dimensions (natural environment, economic environment, facilities and equipment, and physical structure). It is based on the analysis of UPS risks and the driving factors of UPS system resilience.

Enhancing system resilience involves strengthening the adaptive and resistance capacity to coexist with risks. The risk management adaptive cycle covers the “before, during, after, and long-term development” phases (Figure 4). It emphasizes the ability to resist, absorb, adapt, recover, and develop. Enhancing the ex-ante resilience level mainly entails accumulating the emergency relief reserve, risk warning, and disaster management to resist and prevent risks. Enhancing resilience during the event (ex-post) mainly includes disaster response and shortening the period of disaster resistance and adaptation. Enhancing the level of resilience afterward mainly involves recovery and reconstruction of the system and the ability to adapt. The framework of the resilience enhancement strategy for the UPS system is depicted in Figure 5.

### 4.2. Ex-Ante Resilience Enhancement Strategy

Enhancing the resilience strategy of the UPS system revolves around the four-stage framework of ex-ante, in-event, post-event, and long-term development. The following strategy packages contain measures to prevent or mitigate disaster risks.

Preventing and controlling beforehand is a crucial risk management step. The occurrence of UPS safety accidents reflects risk factors breaching the safety protection system and propagating under the single-risk scenario or multi-risk coupling with mutual influence and interaction. The UPS suffers damage and loss with the disruption of overall stability. To reduce risks and enhance resilience for UPS, targeted control and suppression of risk factors should be implemented before events occurrence. The goal is reducing the probability of risk occurrence from the source, thus enhancing system resilience. The proposed actions for the ex-ante resilience enhancement of UPS are expressed in Figure 6.

#### 4.2.1. Enhancing Resilience of the Natural Environment

The identification of the risk sources of UPS due to natural disasters has been widely realized. The top priority after risk recognition is preventing specific risk factors. The measures should be implemented in good time. Firstly, it is necessary to strengthen disaster early warning management, improve the ability to perceive natural-disaster risks and achieve early detection and prevention. After receiving disaster warnings, some urgent actions should forthwith be taken, including preparing emergency resources, protective measures, and other emergency actions. Arrangements should be made promptly to control and evacuate people and minimize impacts on the UPS. 

The hidden dangers should be investigated. Potential risks and secondary accidents must be prevented. The emergency plan for natural disasters should be continually improved in response to changing circumstances, mainly by relevant local government departments. Unlike other urban risks, natural disasters are characterized by inevitable occurrence, not being amenable to elimination, and incurring severe consequences. A holistic plan encompasses the medical, technical, economic, and other particular emergency aspects. The disaster response measures should include orderly public space management and procedures to calm, comfort, and help affected people.

#### 4.2.2. Enhancing Resilience of the Economic Environment

From the economic environment dimension, resilience enhancement refers to adjusting the organization of disaster prevention and mitigation measures from the economic perspective to respond effectively to UPS risks. The government and the private sector have to improve resource allocation to tackle accidents and disasters and response capabilities. The local government and UPS management units have to continually review and upgrade the emergency rescue system and capacity, emergency rescue plans, and prevention and control measures. Adequate funding should be allocated to acquire emergency supplies and reserves and implement rescue operations. It is essential to develop emergency evacuation plans, optimize emergency evacuation routes, and conduct regular evacuation drills. Sound measures to facilitate recovery can be included in a commercial response plan to reduce economic loss.

#### 4.2.3. Enhancing Resilience of Facilities and Equipment

Facilities and equipment are the cornerstone of smooth and safe UPS operation. The complex and inter-related components per se can become the risk sources. Faulty, defective, worn-out, aging, and poorly maintained equipment items could incur safety accidents. As such, they are vulnerable to impacts and failures induced by natural disasters, human-made disasters, terrorist attacks, and other safety incidents. It is necessary to strengthen the upkeep and protection of facilities and equipment to minimize breakdowns and associated losses. Their resilience is related to physical location, exposure, and vulnerability to impacts in the underground milieu. They should be installed at places that can be shielded from disaster impingements and include defensive or protective means to withstand or tolerate risk impacts. In addition, standardizing the operation and maintenance of facility and equipment operators could contribute to risk control. 

#### 4.2.4. Enhancing Resilience of the Physical Structure

A stable and solid underground building structure can significantly reduce the impacts of external forces on the UPS, thus reducing the consequences of impacts. The building structure and space can be designed or modified to resist or alleviate accident or disaster impacts. The capacity and network layout of the passages could be rationalized for normal use and emergency evacuation. The 3D spatial configuration of horizontal and vertical flow paths and floor connections can be evaluated in detail in devising the rescue and evacuation plans. It is pertinent to ensure adequate and effective emergency shelters, passages, and exits. 

### 4.3. Ex-Post Resilience Enhancement Strategy

In-event control constitutes the core of risk management. In-event resilience enhancement can shorten the impact duration of the risk event and reduce losses. Risk occurrence can be divided into five states: optimal, basic operable, recoverable, can-be-mitigated, and failure [43], as depicted in Figure 7. If the risk factors have breached the safety system protection of UPS, targeted control measures must be taken to stop the risk spread, shorten the risk event, and suppress the risk from reaching the failure state to reduce human, property, and economic losses. 

The development of the risk should be accurately judged, and the risk source should be ascertained. The risk must be intervened or blocked to reduce the speed of risk development and contain the risk propagation. The risk source must be stopped from coupling with other potential risk factors, which may aggravate the harmful impacts. 

Second, once an emergency event occurs, a swift response mechanism is needed. For example, accidents and disasters can lead to blocked passages in the UPS. If the number of usable emergency exits is inadequate, it could incur a chain of disastrous congestion, pile-up, and stampede accidents. Therefore, it is necessary to mobilize rapidly alternative emergency rescue channels. The rescue materials should be made available at appropriate locations or conveyed to the incident sites in time. The crisis plan should allow speedy deployment of emergency personnel, evacuation of the disaster site, and dispatch of ambulances, doctors, and other medical and health services. Meanwhile, the social work organizations should be well prepared and primed to provide shelter and sustenance when necessary.

### 4.4. Post-Event Resilience Enhancement Strategy

The UPS post-incident resilience mainly includes space recovery and reconstruction. Recovery refers to the return to normal operation. Reconstruction refers to repairing or replacing structures, facilities, and equipment damaged by accident, to be accomplished in the shortest possible time after effectively controlling the incident and the site. A key goal in the post-event phase is eliminating the root cause of the accident backlash and forestalling the lingering negative impacts.

#### 4.4.1. Restoration and Reconstruction of Facilities and Equipment

The restoration and reconstruction of the material aspects of UPS allow resumption of regular operation and productive activities. Damaged structures and physical facilities can be repaired or rebuilt expeditiously. They may include power, water, drainage, escalators, elevators, gates, etc. If appropriate, the reprovisioned items should be rendered more resistant to damage. The ability of all facilities and equipment to cope with risks could be holistically reviewed to assess the ability to resist risks, with more attention to the more seriously affected areas. The ability to evacuate and respond could be strengthened in a targeted manner.

#### 4.4.2. Recovery and Reconstruction of the Economic Environment

Safety accidents in UPS often result in economic impacts such as damage to facilities, disruption and suspension of rail transport service, and closure of underground businesses. Many resources, such as medical and health, rescue materials, and disaster relief funds, would have been consumed. The priority is restoring underground commercial activities and the operation of underground transportation. The disaster relief funds should be carefully managed and disbursed to maximize their utilities and relief functions. Scientific evaluation of disaster relief resources and emergency plans can identify the shortcomings and inform improvement and effectiveness.

#### 4.4.3. Rehabilitation and Reconstruction of the Social Environment

Safety accidents in UPS, especially those causing heavy casualties, often trigger psychological trauma in victims and reduce people’s general sense of security in the UPS. Therefore, in the aftermath, psychological rehabilitation care and support can provide relief to distressed people and restore their mental health. To enhance users’ understanding of the safe, harmonious, and stable UPS environment, safety education, awareness, and literacy can be transmitted to passengers, customers, managers, and transport staff. A public education campaign can disseminate UPS safety knowledge and emergency measures to reduce casualties and losses. 

#### 4.4.4. Long-Term Development and Enhancement Strategy

To prevent the recurrence of similar accidents and contain the losses, it is necessary to review the disaster’s causes, losses, and rescue procedures to enhance UPS system resilience. The administration should evaluate the technical, economic, environmental, and personnel domains, learn from event experiences, and distill improvement measures. In particular, for extraordinary risk incidents, more effective prevention and protection strategies should be devised. The training of emergency rescue teams and the provision of rescue resources can be strengthened. Public safety education and social awareness of UPS risks and countering measures could be stepped up and more assiduously implemented to reach more citizens. The science and technology of UPS safety and resilience could be more actively studied, and the research findings could be translated into enhanced practices. Information technology can be judiciously enlisted to improve risk monitoring, advocacy, and management.

## 5. Conclusions

Studies of risk resilience of cities in China and abroad are focused on the aboveground realm. This project was conceived to fill the knowledge gap, the bottleneck in UPS safety governance, by applying the relevant resilience concepts and practice. We aimed to develop a new approach to enhance the risk management of the increasingly patronized and expanding urban UPS.

The UPS system was analyzed by the 17 selected resilience indicators in the four dimensions of latent factors: natural environment, facilities and equipment, economic environment, and physical structure. Our primary data were obtained by a questionnaire tailor-made to match the research objectives. Computational software of SPSS and AMOS were employed to construct the structural equation models, test the model fitting, and evaluate model validity and applicability.

The results showed that the economic environment and physical structure factors influenced UPS system resilience more than natural environment and facilities and equipment factors. The computed relative importance of the resilience indicators allowed the development of a package of resilience enhancement measures. The resilience enhancement strategy of the UPS system was developed by combining our results with the findings of cognate risk assessments. The practical recommendations could be realized in four stages: before, during, after, and long-term development, under the four dimensions of latent factors. The improved strategy can suppress impacts during the disaster event, facilitate rescue and evacuation operations, reduce casualties and losses, and permit prompt recovery after the event. Thus, it can enhance the practice of UPS disaster response and provide theoretical guidance for risk control and management in China.

However, limitations also exist in this study. First, the selection of the UPS resilience indicators could be expanded to include more factors to strengthen the predictive power and utility of the model. For example, public safety education, related social awareness, social environment, ecological environment, institutions, spatial structure and other indicators to continue the study. Second, the criteria and questionnaire data selected for this study are from China, so the findings are only applicable to the Chinese region, and further investigation of other regions in future studies is recommended to apply to a wider area.

## Figures and Tables

**Figure 1 ijerph-19-15897-f001:**
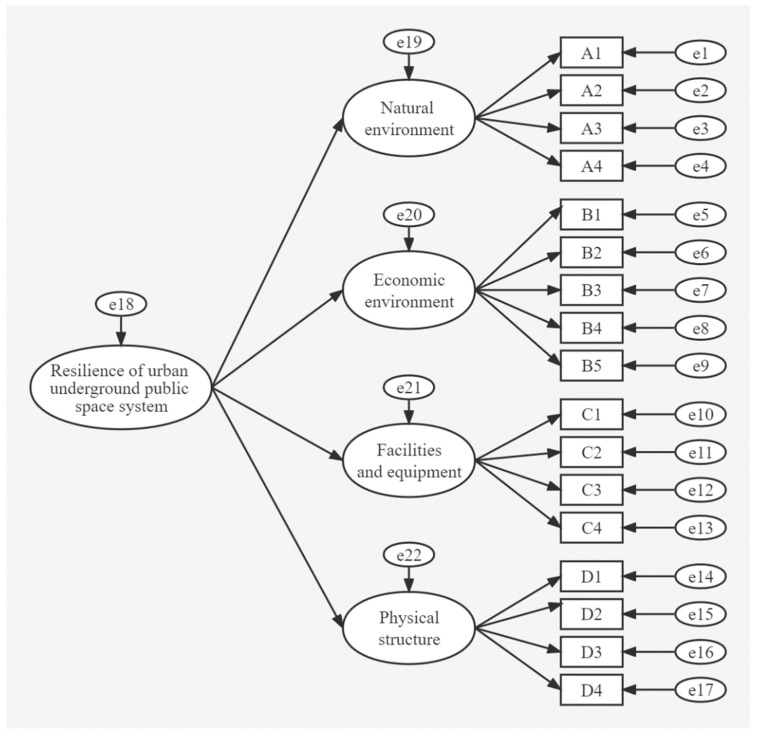
The path diagram of the structural equation model to represent the resilience of the underground public space (UPS) system.

**Figure 2 ijerph-19-15897-f002:**
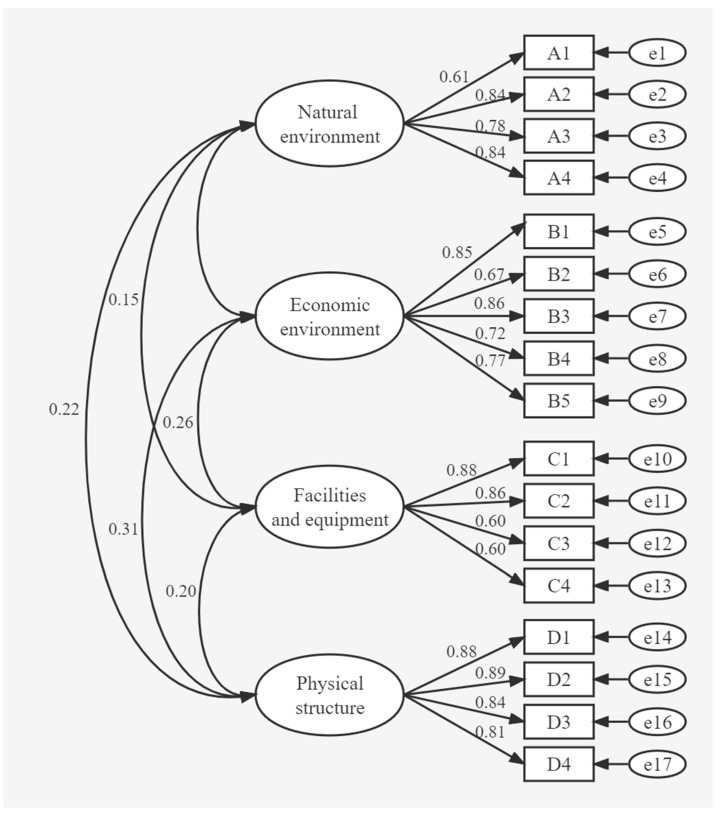
Path coefficients of the first-order structural equation model of underground public space (UPS).

**Figure 3 ijerph-19-15897-f003:**
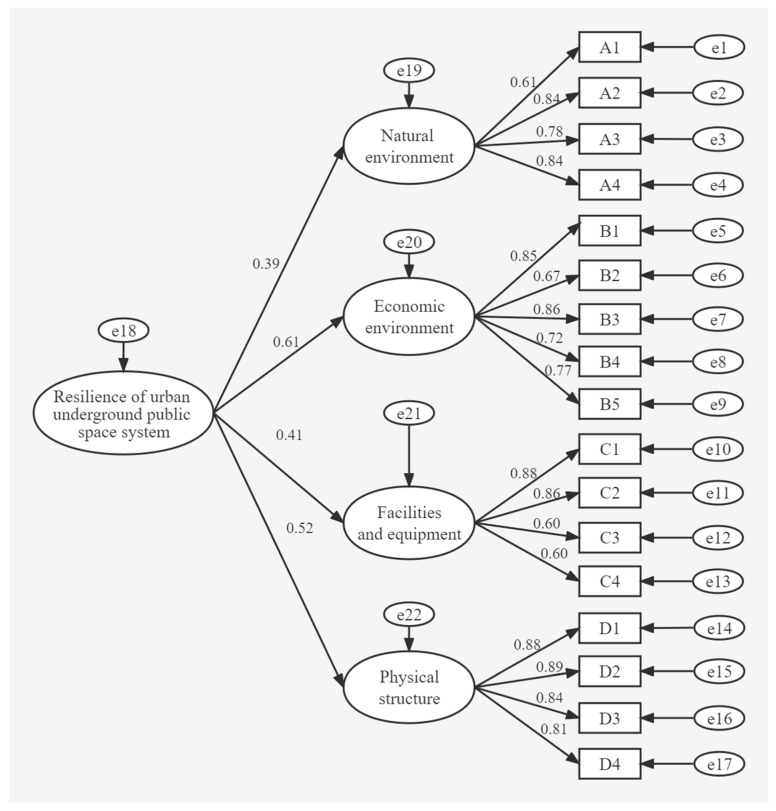
Path coefficients of the second-order structural equations model of underground public space (UPS).

**Figure 4 ijerph-19-15897-f004:**
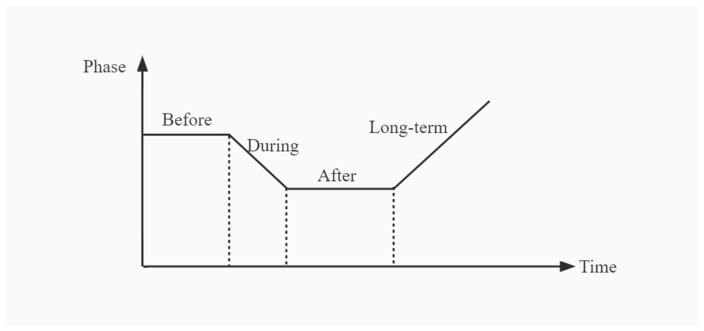
The four phases of the resilience adaptive cycle of the underground public space (UPS) system.

**Figure 5 ijerph-19-15897-f005:**
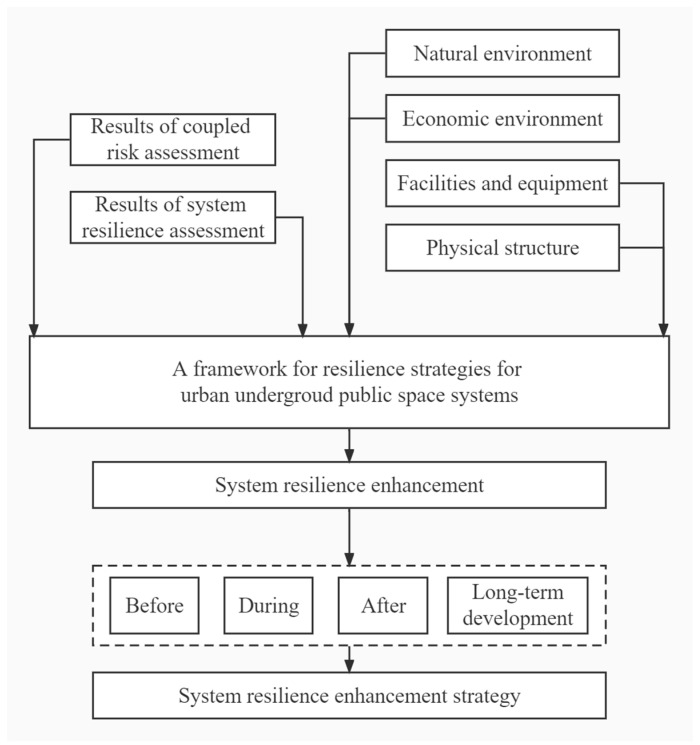
The proposed resilience enhancement strategy for the underground public space (UPS) system.

**Figure 6 ijerph-19-15897-f006:**
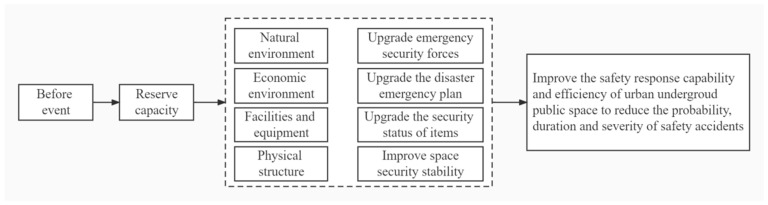
The proposed actions to improve the ex-ante resilience strategy for the underground public space (UPS) system.

**Figure 7 ijerph-19-15897-f007:**
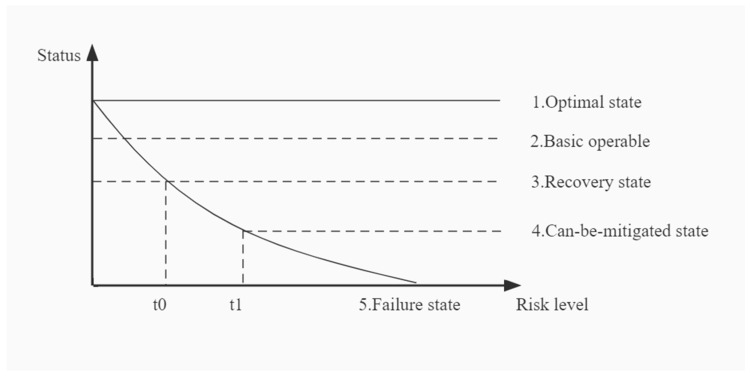
The five key states of accident risk occurrence.

**Table 1 ijerph-19-15897-t001:** The four latent factors and their resilience indicators of the underground public space (UPS) system.

Latent Factor	Resilience Indicator	Code
Natural environment	Frequency of natural disasters	A1
	Intensity of natural disaster	A2
	Disaster response measures	A3
	Extent of damage caused by the disaster	A4
Economic environment	Quantity of emergency relief materials	B1
	Financial allocation for post-disaster relief	B2
	Material reserves for post-disaster reconstruction	B3
	Number of professional rescue personnel	B4
	Extent to which relief materials can respond to disaster	B5
Facilities and equipment	Electricity supply system	C1
	Fire-fighting facilities	C2
	Ventilation system	C3
	Flood control and drainage system	C4
Physical structure	Number of underground space levels	D1
	Height of underground space	D2
	Number of stairs/escalators	D3
	Number of emergency exits	D4

**Table 2 ijerph-19-15897-t002:** The four research hypothesis statements.

Code	Hypothesis
**1 (H1)**	*Natural environment factors have a positive effect on the resilience of UPS systems*
**2 (H2)**	*Economic environment factors have a positive effect on the resilience of UPS systems*
**3 (H3)**	*Facilities and equipment factors have a positive effect on the resilience of UPS systems*
**4 (H4)**	*Physical environment factors have a positive effect on the resilience of UPS systems*

**Table 3 ijerph-19-15897-t003:** Key socio-economic traits of the questionnaire respondents.

Socio-Economic Trait	Category	Frequency	(%)
Education level	Technical college	43	20.7
	Bachelor degree	67	32.2
	Master degree and above	63	30.3
	Others	35	16.8
Work experience	1–2 years	56	26.9
	3–5 years	79	38.0
	6–10 years	46	22.1
	>10 years	5	2.4
	None	22	10.6
Occupation	Corporation	22	10.6
	Service industry	15	7.2
	Construction	72	34.6
	Transportation	36	17.3
	Education	18	8.7
	Freelance	11	5.3
	Other	8	3.8
	Student	26	12.5

**Table 4 ijerph-19-15897-t004:** Descriptive statistics of importance rating of 17 resilience indicators (sample size = 250, minimum value = 1 for “Very unimportant, and maximum value of 5 for “Very important”).

Resilience Indicator	Average Rating Value	Rank of Average Value	Standard Deviation	Variance
Natural environment:				
Frequency of natural disasters	2.83	14	1.179	1.390
Intensity of natural disaster	2.60	15	1.236	1.527
Disaster response measures	2.53	16	1.307	1.709
Extent of damage caused by the disaster	2.46	17	1.326	1.757
Nominal average	2.61	15.50		
Economic environment:				
Quantity of emergency relief materials	3.55	8	1.299	1.688
Financial allocation for post-disaster relief	3.60	7	1.414	1.999
Material reserves for post-disaster reconstruction	3.64	5	1.340	1.796
Number of professional rescue personnel	3.61	6	1.318	1.737
Extent to which relief materials can respond to disaster	3.41	11	1.425	2.031
Nominal average	3.56	7.40		
Facilities and equipment:				
Electricity supply system	3.65	4	1.277	1.630
Fire-fighting facilities	3.79	1	1.283	1.646
Ventilation system	3.75	2	1.286	1.655
Flood control and drainage system	3.75	3	1.203	1.447
Nominal average	3.74	2.50		
Physical structure:				
Number of underground space layers	3.46	9	1.461	2.134
Height of underground space	3.46	10	1.461	2.134
Number of stairs/escalators	3.32	12	1.402	1.966
Number of emergency exits	3.21	13	1.469	2.158
Nominal average	3.36	11.00		
Overall nominal average	3.33	9.00		

**Table 5 ijerph-19-15897-t005:** Cronbach’s alpha coefficient and bipolar reliability grading criteria.

Range of Cronbach’s Alpha Coefficient	Bipolar Reliability Grade
<0.6	Strongly unreliable
0.6–0.7	Moderately unreliable
0.7–0.8	Neutral
0.8–0.9	Moderately reliable
>0.9	Strongly reliable

**Table 6 ijerph-19-15897-t006:** Cronbach’s alpha output of the reliability test.

Latent Factor	Resilience Indicator	Cronbach Alpha after Deletion of Items	Alpha
Natural environment	Frequency of natural disaster	0.862	0.851
	Intensity of natural disaster	0.783	
	Disaster response measures	0.811	
	Extent of damage caused by the disaster	0.778	
Economic environment	Quantity of emergency relief materials	0.882	0.908
	Financial allocation for post-disaster relief	0.878	
	Material reserves for post-disaster reconstruction	0.902	
	Number of professional rescue personnel	0.883	
	Extent to which relief supplies can respond to disaster	0.893	
Facilities and equipment	Electricity supply system	0.757	0.832
	Fire-fighting facilities	0.764	
	Ventilation system	0.813	
	Flood control and drainage system	0.811	
Physical structure	Number of underground space levels	0.882	0.914
	Height of underground space	0.879	
	Number of stairs/escalators	0.893	
	Number of emergency exits	0.902	

**Table 7 ijerph-19-15897-t007:** KMO value range and significance grading scale.

KMO Value Range	Grading
≥0.9	Excellent
0.8–0.9	Very good
0.7–0.8	Good
0.6–0.7	Moderate
0.5–0.6	Poor
<0.6	Very poor

**Table 8 ijerph-19-15897-t008:** Results of KMO measure and Bartlett’s sphericity test.

KMO Sampling Fitness Measure		0.831
Bartlett’s sphericity test	Approximate cardinality	2076.184
	Degrees of freedom	136
	Significance	0.00 ***

Note: *** represent for sig.value Less than 0.001.

**Table 9 ijerph-19-15897-t009:** Results of the convergent validity test for the questionnaire data.

Indicator Code	Standardized Factor Loading	Average Variance Extracted (AVE)	Combined Reliability (CR)
A1	0.612	0.601	0.856
A2	0.839		
A3	0.782		
A4	0.845		
B1	0.850	0.667	0.909
B2	0.871		
B3	0.857		
B4	0.723		
B5	0.774		
C1	0.885	0.560	0.831
C2	0.858		
C3	0.599		
C4	0.600		
D1	0.877	0.730	0.915
D2	0.887		
D3	0.842		
D4	0.809		

**Table 10 ijerph-19-15897-t010:** Parameter estimates and fitness test of the first-order structural equations model.

Resilience Indicator	Non-Standardized Coefficient Estimate	Standardized Coefficient Estimate	Standard Error (SE)	Combined Reliability (CR)	*p* ^a^
Natural environment–A1	1.000	0.612			
Natural environment–A2	1.436	0.839	0.159	9.038	***
Natural environment–A3	1.416	0.782	0.163	8.682	***
Natural environment–A4	1.551	0.845	0.171	9.068	***
Economic environment–B1	1.000	0.850			
Economic environment–B2	1.115	0.871	0.071	15.804	***
Economic environment–B3	1.039	0.857	0.067	15.397	***
Economic environment–B4	0.862	0.723	0.072	11.908	***
Economic environment–B5	0.998	0.774	0.076	13.150	***
Facilities and equipment–C1	1.000	0.885			
Facilities and equipment–C2	0.974	0.858	0.073	13.353	***
Facilities and equipment–C3	0.682	0.599	0.076	8.985	***
Facilities and equipment–C4	0.639	0.600	0.071	9.010	***
Physical structure–D1	1.000	0.877			
Physical structure–D2	1.012	0.887	0.059	17.206	***
Physical structure–D3	0.922	0.842	0.059	15.730	***
Physical structure–D4	0.928	0.809	0.063	14.688	***

^a^ *** means *p* < 0.001 significance level.

**Table 11 ijerph-19-15897-t011:** Criteria values and results of the model fitting test.

Fitting Index	χ^2^	df	χ^2^/df	NFI	IFI	TLI	CFI	GFI	RMSEA
Fitting results	154.176	113	1.364	0.928	0.980	0.975	0.979	0.923	0.042
Threshold criteria ^a^	/	/	<3.0	>0.9	>0.9	>0.9	>0.9	>0.9	<0.05

^a^ Reference sources for the threshold criteria: [41].

**Table 12 ijerph-19-15897-t012:** Parameter estimates and fitness test of the second-order structural equation model.

Resilience Indicator	Non-Standardized Coefficient Estimate	Standardized Coefficient Estimate	Standard Error (SE)	Combined Reliability (CR)	*p* ^a^
Resilience of underground public space system–Natural Environment	0.282	0.392	0.078	3.604	***
Resilience of underground public space system–Economic environment	0.676	0.613	0.130	5.217	***
Resilience of underground public space system–Facilities and equipment	0.457	0.406	0.116	3.932	***
Resilience of underground public space system–Physical structure	0.664	0.520	0.138	4.800	***
Natural environment–A1	1.000	0.612			
Natural environment–A2	1.436	0.839	0.159	9.038	***
Natural environment–A3	1.416	0.782	0.163	8.682	***
Natural environment–A4	1.551	0.845	0.171	9.068	***
Economic environment–B1	1.000	0.850			
Economic environment–B2	1.115	0.871	0.071	15.804	***
Economic environment–B3	1.039	0.857	0.067	15.397	***
Economic environment–B4	0.862	0.723	0.072	11.908	***
Economic environment–B5	0.998	0.774	0.076	13.15	***
Facilities and equipment–C1	1.000	0.885			
Facilities and equipment–C2	0.974	0.858	0.073	13.353	***
Facilities and equipment–C3	0.682	0.599	0.076	8.985	***
Facilities and equipment–C4	0.639	0.600	0.071	9.010	***
Physical structure–D1	1.000	0.877			
Physical structure–D2	1.012	0.887	0.059	17.206	***
Physical structure–D3	0.922	0.842	0.059	15.730	***
Physical structure–D4	0.928	0.809	0.063	14.688	***

^a^ *** means *p* < 0.001 significance level.

**Table 13 ijerph-19-15897-t013:** The overall fitness of the second-order structural equation model.

Fitting Index	X^2^	df	X^2^/df	NFI	IFI	TLI	CFI	GFI	RMSEA
Fitting value	154.385	115	1.342	0.928	0.981	0.977	0.980	0.923	0.041
Threshold criteria ^a^	/	/	<3.0	>0.9	>0.9	>0.9	>0.9	>0.9	<0.05

^a^ Reference sources for the threshold criteria: [41].

**Table 14 ijerph-19-15897-t014:** Path coefficients and normalized weights of latent factors and resilience indicators.

Latent Factor	Path Coefficient	Normalized Weight	Resilience Indicator	Path Coefficient	Normalized Weight	Total Normalized Weight
Natural environment	0.392	0.203	Frequency of natural disasters	0.612	0.199	0.040
			Intensity of natural disaster	0.839	0.273	0.055
			Disaster response measures	0.782	0.254	0.052
			Extent of damage caused by the disaster	0.845	0.275	0.056
Economic environment	0.613	0.317	Quantity of emergency relief materials	0.850	0.209	0.066
			Financial allocation for post-disaster relief	0.871	0.214	0.068
			Material reserves for post-disaster reconstruction	0.857	0.210	0.067
			Number of professional rescue personnel	0.723	0.177	0.056
			Extent to which relief materials can respond to disasters	0.774	0.190	0.060
Facilities and equipment	0.406	0.210	Electricity supply system	0.885	0.301	0.063
			Fire-fighting facilities	0.858	0.292	0.061
			Ventilation system	0.599	0.204	0.043
			Flood control and drainage system	0.600	0.204	0.043
Physical structure	0.520	0.269	Number of underground space levels	0.877	0.257	0.069
			Height of underground space	0.887	0.260	0.070
			Number of stairs/escalators	0.842	0.247	0.066
			Number of emergency exits	0.809	0.237	0.064

## Data Availability

Not applicable.

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
