# Peer review of "Research on Risk and Resilience Evaluation of Urban Underground Public Space"

_ijerph, 2022, doi:10.3390/ijerph192315897_

Round 1

Reviewer 1 Report

Very interesting topic, well developed, highly innovative and well written. The research is mainly focusing on underground urban public space, with a rather weak connection to the broader surrounding environment. With a more elaborated connection to the broader environment, the scientific approach could have been a further and more specialised development of the ARUP cycle approach to resilience policies, and the 100 resilient cities programme, with stronger policy implications. Also, the latent factors and resilience indicators used to shape up the questionnaire could gain in reliability if, instead of being predefined, they were the outcomes of a more extensive public consultation -such as the living labs. Finally, I would emphasize the part of the conclusions stating that there should be further in-depth investigations to the selection of UPS resilience indicators, concerning  social environment, ecological environment, institutions, spatial structure, legal and institutional framework etc., since public safety education and related social awareness are among the few, and most important notions being able to cover resilence in UPS, in a generalized, integrated approach.

Author Response

Reviewer1

  • Very interesting topic, well developed, highly innovative and well written. The research is mainly focusing on underground urban public space, with a rather weak connection to the broader surrounding environment. With a more elaborated connection to the broader environment, the scientific approach could have been a further and more specialised development of the ARUP cycle approach to resilience policies, and the 100 resilient cities programme, with stronger policy implications.

Response: This study focuses on the risks that can occur in the internal environment of UPS, and considers the study of risk resilience under internal operational mechanisms. The weak linkage with the surrounding environment is also a limitation of this study, and future studies can strengthen this part of the linkage to provide assistance to urban resilience studies.

  • Also, the latent factors and resilience indicators used to shape up the questionnaire could gain in reliability if, instead of being predefined, they were the outcomes of a more extensive public consultation -such as the living labs.

Response: In the preliminary questionnaire design, we have included living laboratories in the target range of the questionnaire, which also reflects the results of the broader public consultation.

  • Finally, I would emphasize the part of the conclusions stating that there should be further in-depth investigations to the selection of UPS resilience indicators, concerning  social environment, ecological environment, institutions, spatial structure, legal and institutional framework etc., since public safety education and related social awareness are among the few, and most important notions being able to cover resilence in UPS, in a generalized, integrated approach.

Response:Indeed, public safety education and related social awareness are among the few resilience indicators that best cover UPS, and we have added these two indicators to the limitations of the study in the hope that future studies can start from there.

Reviewer 2 Report

The research was on the evaluation of risk and resilience in public urban underground space. The topic is important. References are up-to-date. There are some items for authors to consider for improving the quality of the manuscript:

1. The research aim and objectives were missing.

2. Missing (4) in Line 151.

3. What country is the document of "Design Code for the Internal Environment of Urban Underground Space" based on? Without making clear of this information, factor analysis is meaningless because different countries have different situations.  

4. In Table 4, it indicated the sample size=280, which is different from what is stated in Line 189.

5. The research limitations were missing.

Author Response

Reviewer2

The research was on the evaluation of risk and resilience in public urban underground space. The topic is important. References are up-to-date. There are some items for authors to consider for improving the quality of the manuscript:

  1. The research aim and objectives were missing.

Response: The research aim and objectives have been added to the first part of the article. (see page 3 lines 100-102)

  1. Missing (4) in Line 151.

Response: Thanks for your suggestion. Missing (4) has been added.

  1. What country is the document of "Design Code for the Internal Environment of Urban Underground Space" based on? Without making clear of this information, factor analysis is meaningless because different countries have different situations.  

Response: The document "Design Code for the Internal Environment of Urban Underground Spaces" is based on China and is numbered CECS 441:2016.

  1. In Table 4, it indicated the sample size=280, which is different from what is stated in Line 189.

Response: After verification, the sample size was 250 and the data in Table 4 was corrected.

  1. The research limitations were missing.

Response:The research limitations have been added in Section 5 of the article.(see page 20 lines 537-544)